# Telomere Length Is Correlated with Resting Metabolic Rate and Aerobic Capacity in Women: A Cross-Sectional Study

**DOI:** 10.3390/ijms232113336

**Published:** 2022-11-01

**Authors:** Rujira Nonsa-ard, Ploypailin Aneknan, Terdthai Tong-un, Sittisak Honsawek, Chanvit Leelayuwat, Naruemon Leelayuwat

**Affiliations:** 1Biomedical Sciences Program, Graduate School, Khon Kaen University, Khon Kaen 40002, Thailand; 2Exercise and Sport Sciences Development and Research Group, Khon Kaen University, Khon Kaen 40002, Thailand; 3Department of Physiology, Faculty of Medicine, Khon Kaen University, Khon Kaen 40002, Thailand; 4Centre of Excellence in Osteoarthritis and Musculoskeleton, Faculty of Medicine, Chulalongkorn University, Bangkok 10330, Thailand; 5Centre for Research and Development of Medical Diagnostic Laboratories (CMDL), Faculty of Associated Medical Sciences, Khon Kaen University, Khon Kaen 40002, Thailand; 6Graduate School, Khon Kaen University, Khon Kaen 40002, Thailand

**Keywords:** telomere length, metabolism, maximum oxygen consumption, vitamin C, high-sensitivity C-reactive protein

## Abstract

This study investigated the associations between relative telomere length (RTL) and resting metabolic rate (RMR), resting fat oxidation (RFO), and aerobic capacity and whether oxidative stress and inflammation are the underlying mechanisms in sedentary women. We also aimed to determine whether the correlations depend on age and obesity. Sixty-eight normal weight and 66 obese women participated in this study. After adjustment for age, energy expenditure, energy intake, and education level, the RTL of all participants was negatively correlated with absolute RMR (RMR_AB_) and serum high-sensitivity C-reactive protein (hsCRP) concentration, and positively correlated with maximum oxygen consumption ($\dot{V}$O_2max_) (all *p* < 0.05). After additional adjustment for adiposity indices and fat-free mass (FFM), RTL was positively correlated with plasma vitamin C concentration (*p* < 0.05). Furthermore, after adjustment for fasting blood glucose concentration, RTL was negatively correlated with age and positively correlated with $\dot{V}$O_2max_ (mL/kg FFM/min). We found that normal weight women had longer RTL than obese women (*p* < 0.001). We suggest that RTL is negatively correlated with RMR_AB_ and positively correlated with aerobic capacity, possibly via antioxidant and anti-inflammatory mechanisms. Furthermore, age and obesity influenced the associations. We provide useful information for the management of promotion strategies for health-related physical fitness in women.

## 1. Introduction

Telomeres are nucleoprotein complexes located at the ends of eukaryotic chromosomes [1,2]. Telomeres play a role in protecting chromosome stability and integrity. The length of telomeres naturally shortens during cell replication during ageing [2]. Thus, telomere length (TL) is commonly used as a biomarker of biological ageing [3] and age-related chronic metabolic diseases, including obesity [4,5,6].

Worldwide, obesity is a leading risk factor for morbidity and mortality [7]. Thus, it is worth exploring genetic factors correlated with obesity to develop an intervention for this condition. Obese individuals have been shown to have high fat mass (FM), skeletal muscle mass, and fat-free mass (FFM), which are important factors correlated with the resting metabolic rate (RMR) [8,9]. The RMR is the minimum energy that is required to maintain the essential homeostasis of life. It accounts for the majority (60–70%) of daily energy expenditure [10]. Therefore, RMR change influences body homeostasis and thus impacts health status. However, no study has been conducted to explore an association between TL and RMR in obese individuals.

In addition, fat storage in obese individuals leads to increased circulating free fatty acid availability for muscle uptake. In addition to the increased circulating FFA availability, the upregulated FFA transporter FAT/CD36 [11] resulted in increased FFA uptake into subcutaneous and visceral adipose tissues in obese participants. These factors were shown to enhance the resting fat oxidation (RFO) rate, which was reported to increase in obese individuals [12]. In fact, the RFO is also a powerful predictor of biological ageing [13]. Thus, RFO should be correlated with TL in obese individuals. Nevertheless, no evidence has shown the association of RFO with TL in obese people.

Importantly, obesity also impairs physical performance, especially aerobic capacity, leading to an increased mortality rate with increasing age [14]. This evidence was genetically supported by a previous study reporting the association of aerobic capacity with TL [15,16]. Previous studies have shown a significant association in various populations, including young and middle-aged (20–49 years) [16] adults and older men (55–72 years) [15]. It is important to note that those mentioned studies were performed in male nonobese subjects. Although the study by Loprinzi (2015) was performed in both sexes, it collected data on only young to middle-age, nonobese individuals. Therefore, more research in samples with a wide range of ages should be performed to clarify the association of TL and aerobic capacity in obese women.

Obesity is characterized by increased oxidative stress and the release of inflammatory cytokines [17], which reduce TL [18,19,20]. A G-rich strand, the structure of the telomere, is susceptible to the action of oxidants, leading to an acceleration in the rate of shortening [21,22]. Moreover, chronic inflammation caused by oxidative damage to DNA was shown to influence TL attrition [18]. Thus, obesity is reported to accelerate telomere attrition [3,23]. However, it is less clear whether the mechanisms of oxidative stress and inflammation can explain the associations of TL attrition with RMR, RFO, and aerobic capacity in healthy sedentary obese women. Therefore, to define a mechanism through which TL attrition relates to RMR, RFO, and aerobic capacity in obese individuals, oxidative stress and inflammation have been assessed.

The present study was conducted to examine the associations of RTL with RMR, RFO, and aerobic capacity in Thai women and to determine whether oxidative stress and inflammation are involved in the underlying mechanisms of the associations. We further explore whether the associations depend on age and obesity. We hypothesized that RTL is correlated with RMR, RFO, aerobic capacity, age, and markers of obesity in sedentary healthy women via mechanisms of oxidative stress and inflammation. We also hypothesized that there were differences in the associations between women with obesity and normal weight in each age subgroup.

## 2. Results

Of 163 women, 134 women (68 NW and 66 OB) were recruited. Twenty-nine women (14 NW and 15 OB) were excluded because they did not meet the inclusion criteria. Each of four age subgroups (20–29-, 40–49-, and 50–59-year-old) comprised 34 participants (17 NW, 17 OB). Only the 30–39-year-old group was composed of 32 participants (17 NW and 15 OB) because of the COVID-19 situation. Obese participants had been obese for longer than three years.

### 2.1. Demographic and Physiological Characteristics

As expected, the OB group had significantly greater adiposity indices, FFM, systolic blood pressure, FBG, HbA1C, HOMA-IR, HbA1C, triacylglycerol (TG), and EI, EE, and lower high-density lipoprotein cholesterol (HDL-c) than the NW group (Appendix A). All adiposity indices of the OB group were greater than those of the NW group in all age subgroups (all *p* < 0.001) (Appendix A). The RTL was found to be longer in NW than OB (1.1 ± 0.3 in NW and 0.8 ± 0.2 in OB, *p* < 0.001), as shown in Figure 1.

### 2.2. RMR, RFO, Aerobic Capacity, Oxidative Stress, Inflammation, and RTL

The OB group had significantly greater absolute RMR (RMR_AB_) (kcal/day) and RFO (RFO_AB_) (g/min) values than the NW group (both *p* < 0.01). RMR relative to BM (RMR_BM_) and relative to FFM (RMR_FFM_) and RFO relative to BM (RFO_BM_) were higher in the NW group than in the OB group (all *p* < 0.05) (Table 1). However, RFO relative to FFM (RFO_FFM_) was not significantly different between groups (*p* = 0.880). Both $\dot{V}$O_2max_ relative to BM (mL/kg BM/min) and to FFM (mL/kg FFM/min) were lower in the OB group than in the NW group (both *p* < 0.05). Plasma MDA (µmol/mL), plasma vitamin C (µmol/L), and serum hsCRP (mg/L) concentrations were greater in the OB group than in the NW group (all *p* < 0.001). The RTL of the OB group was shorter than that of the NW group (*p* < 0.001). At least two age subgroups, RMR_AB_, plasma vitamin C, plasma MDA, serum hsCRP were greater and RMR_BM_, RMR_FFM_, $\dot{V}$O_2max_, and RTL were lower in OB than NW group. Whereas RFOs were different between groups at highest one age subgroup (Appendix A).

### 2.3. Heatmaps of Raw Data and Correlations between RTL and All the Variables

We generated a heatmap of the raw data (normalized to mean or median) and examined the correlation between all the outcomes in the overall, NW, and OB groups (Figure 2). In the overall data, the correlation between RTL and age (*p* < 0.05), adiposity indices (*p* < 0.001), FFM (*p* < 0.01), RMR_AB_ (*p* < 0.05), and hsCRP (*p* < 0.01) showed significant negative correlations, while the correlation between RTL and $\dot{V}$O_2max_ was positive (*p* < 0.05). The NW group showed a significant negative correlation between RTL and age and RMR_FFM_, but there were positive correlations between the FFM and plasma vitamin C. However, the OB group did not show any significant correlations between RTL and all the variables. Furthermore, heatmaps of the full correlation matrix of all the data are shown in Figure 1.

In the age subgroups, the heatmaps of the raw data (normalized to mean or median) and correlations showed results that were opposite to those from the all-age data. The NW group did not have any significant correlation between RTL and all variables in any age subgroup (Figure 3). However, the OB group had significant correlations between RTL and many variables. The OB group had a negative correlation in RMR_FFM_ and $\dot{V}$O_2max_ adjusted for BM and FFM among the 20-29-year-old but had a positive correlation in RMR_FFM_ and $\dot{V}$O_2max_ adjusted for FFM among the 40-59-year-old (Figure 3).

### 2.4. Correlation between RTL and All Adjusted Variables

#### 2.4.1. RMR, RFO, and Aerobic Capacity

After adjustment for age, EE, EI, and EL (model 1), the overall data showed negative associations between RTL and RMR_AB_ (*p* = 0.006). However, after additional adjustment for BMI, W, body fat (%), and FFM (model 2), a negative association between RTL and RMR_AB_ was found in the NW group (*p* = 0.049). Moreover, RTL was negatively correlated with RMR_BM_ and RMR_FFM_ in the NW group (models 1 and 2) (all *p* < 0.05). After adjustment for FBG (model 3), a negative association was found only between RTL and RMR_FFM_ in NW group (*p* = 0.027) (Table 2).

Overall, RTL was positively correlated with $\dot{V}$O_2max_ adjusted for BM (models 1 and 3) and FFM (model 3) (all *p* < 0.05) (Table 2). In contrast, RTL was not found to be significantly correlated with either absolute or relative RFO values in all participants.

#### 2.4.2. Association between RTL and Oxidative Stress and Inflammation

After adjustment, RTL was not found to be correlated with plasma MDA in either the NW or OB groups. Furthermore, RTL had a positive association with plasma vitamin C (model 2; *p* = 0.049) in all participants. In addition, RTL was found to be positively correlated with plasma vitamin C (model 3) in the NW and OB groups (model 4; adjusted for HOMA-IR) (both *p* < 0.05). Furthermore, in all participants, RTL was negatively correlated with serum hsCRP concentration (models 1 and 3; both *p* < 0.01) (Table 2).

#### 2.4.3. Associations between RTL and Age, Anthropometry, and Body Composition

Overall, RTL was inversely correlated with age (model 3; *p* = 0.02), BMI, W, BF (%), and FFM (models 1 and 3; all *p* < 0.01). However, in the NW group, RTL was negatively correlated with age (after adjustment for EI, EE, EL and model 3; both *p* < 0.05) and positively correlated with FFM (model 3; *p* = 0.004). However, in the OB group, RTL was not correlated with any outcomes (Table 2).

### 2.5. Associations between RTL and RMR, RFO, Aerobic Capacity, Oxidative Stress, and Inflammation in Age Subgroups

Overall, after being adjusted for EE, EI, and EL, RTL was inversely correlated with adiposity indices, FFM, RMR_BM_, RFO_AB_, and serum hsCRP in mostly 20–39-year-old (all *p* < 0.05) (Appendix A).

In the NW group, after being adjusted for age, EE, EI, and EL, RTL was negatively correlated with RMR_FFM_ (model 4 in 30–39-year-old; *p* < 0.05) and positively association with $\dot{V}$O_2max_ adjusted for FBG (model 3 in 30–39-year-old; *p* < 0.05). In addition, RTL was positively correlated with plasma vitamin C (model 3 in 50–59-year-old; *p* = 0.033) (Appendix A).

In the OB group after adjusted for age, EE, EI, and EL, RTL was positively correlated with RMR_AB_ (model 2; *p* = 0.017), RMR_BM_ (models 1 and 2 in 40–49-year-old, both *p* < 0.05), RMR_FFM_ (models 2 and 3 in 40–49-year-old, both *p* < 0.05) and negatively correlated with RMR_FFM_ (model 1 in 20–29-year-old; *p* < 0.05) (Appendix A). Furthermore, RTL was negatively correlated with $\dot{V}$O_2max_ relative to BM (models 1 and 2 in 20–29-year-old) and to FFM (models 1, 2, and 5 in 20–29-year-old, but found a positive association in models 1 and 2 in 40–49-year-old, and model 5 in 50–59-year-old) (all *p* < 0.05). In addition, RTL was positively correlated with plasma vitamin C (model 3 in 50–59-year-old; *p* = 0.05) (Appendix A).

## 3. Discussion

In this study, RTL in 20- to 59-year-old women had a negative association with RMR_AB_ and a positive association with aerobic capacity in overall sedentary women. However, no significant association between RTL and RFO was found. Moreover, separate data according to obesity status both in the overall sample and age subgroups showed different results. This may confirm that the associations are dependent on age and obesity. Interestingly, the mechanism underlying the longer RTL may be explained by the antioxidant and anti-inflammatory effects. Furthermore, we confirmed the findings of previous studies that RTL attrition was correlated with increasing age, obesity, and FFM in sedentary women.

To our knowledge, this is the first study showing that in sedentary women, RTL was inversely correlated with RMR_AB_ (kcal/day) after adjustment for age, EI, EE, and EL. However, after additional adjustment for adiposity indices (BMI, W, and % BF) and FFM, the relationship disappeared. This evidence reflected that adiposity indices and FFM are potential factors affecting the associations. This finding is comparable to our results in this study showing the association between RTL and adiposity indices (BMI, W, % BF) and FFM.

Interestingly, regarding the association of RTL with aerobic capacity, we are also the first to find lower RTL attrition with increasing aerobic capacity indicated by $\dot{V}$O_2max_ adjusted for BM and FFM after adjustment for age, EI, EE, and EL, and FBG in an overall sample of women. Our data are consistent with data from male participants in previous studies [15,16]. In fact, the relationship was also recently found in women, but their aerobic capacity was indicated by the six-minute walk test (6MWT) [24]. The 6MWT is less accurate than the expired air sample collected during the graded exercise test used in this study. As aforementioned, the findings support our hypothesis on the associations between TL and RMR and aerobic capacity.

Unexpectedly, in the association of RTL with RFO in 20- to 59-year-old, we did not find any significant results. This may be due to a wide range of standard deviations of RFO. The other reason is due to a small number of age subgroups that had differences regarding obesity status, i.e., RFO_AB_ was only different in one age subgroup (30–39-year-old). However, in RFO_FFM,_ there was no age subgroup that exhibited the difference.

In the overall sample of women, we found an association of RTL with RMR_AB_ and RMR_BM_ in younger women (20–39 and 20–29-year-old, respectively). However, RTL was not correlated with RMR_FFM_. This also showed the impact of FFM on the associations, and we found an association of RTL with RFO_AB_ in the middle age group (30–39-year-old). Of note, our results that revealed an association only in the 30–39-year-old age group do not confirm that RFO is a powerful genetic predictor of biological ageing, as suggested by Amaro-Gahete (2019) [13]. The explanation may be due to the small number of age subgroups. The age subgroup analysis is our limitation since we did not calculate the sample size for this analysis. Therefore, a larger cohort for the age subgroup analysis is necessary.

Furthermore, our results on the differences between participants in the association with adiposity indices support our hypothesis. Overall, the NW group showed a negative relationship of RTL with age after adjustment for FBG and with RMR_FFM_ after adjustment for the variables in model 1 and a positive relationship with FFM and plasma vitamin C after adjustment for FBG, whereas the OB group showed a positive relationship only with plasma vitamin C after adjustment for HOMA-IR. In contrast, regarding the age subgroup data, in the OB group RTL showed relationships with more variables, including FFM (model 4), RMR (models 1 and 2), and $\dot{V}$O_2max_ (mL/kg FFM/min) (models 1, 2, and 5). Most associations in the OB group were found in individuals with an older age (40–59-year-old), but associations in the NW group were found in individuals with a younger age (20–39-year-old). We do not have a clear explanation for this finding. The fact that metabolic disorders are mostly occurred at an older age may result in associations in individuals with an older age in the OB group. Of note, the positive relationship of RTL with W in NW and with RMR in OB seems surprising. A larger cohort study should be conducted to clarify these data.

Regarding the mechanism of the association of RTL, the data do not fully support our hypothesis. We found only a positive association between RTL and vitamin C concentration and a negative association with serum hsCRP concentration. The associations between RTL and plasma vitamin C and serum hsCRP concentrations in all women in this study were consistent with previous studies [25,26]. Moreover, we found an association of RTL with plasma vitamin C concentration in the oldest age subgroup in both the NW and OB groups. This may confirm the importance of antioxidants in preventing RTL attrition. However, we did not find associations with serum hsCRP or plasma MDA concentrations in either group. We may need a larger sample size in the age subgroups to explore the significant association.

Additionally, we confirmed the inverse association of RTL with age in all women only after adjustment for FBG. Krasnienkov and coworkers (2018) also found this significant association after adjustment for FBG [27]. These findings are consistent with previous studies that showed RTL attrition with increasing age in a cross-sectional study in women 40–60-year-old [28] and obese women 18–76-year-old [29]. In addition, we found that women with obesity had shorter RTL than their normal weight counterparts. This could be due to a long period of the obesity since the OB participants have been obese for longer than three years.

This study has several limitations, including the clinical condition. Our participants had no underlying diseases. Therefore, our results may not be applicable to women with clinical disorders. Additionally, our results could not be applicable to men because we recruited only female participants. Furthermore, to improve the accuracy of the results, body composition should be measured using dual-energy X-ray absorptiometry (DXA) or magnetic resonance imaging (MRI). We suggest performing further studies that collect data from more participants older than 65 years, including male participants, and measuring oxidative damage at the DNA level. Furthermore, a further study examining the role of hormones on the associations of RTL with the variables should be performed.

## 4. Methods and Materials

### 4.1. Subjects

Our sample size was 136 (including 20% dropout rate) calculated by G*Power 3.1 [30] women and based on a previous study [3] with power = 0.80, α = 0.05, and effect size = 0.18 [31]. We divided the participants into two groups: normal weight (NW) (*n* = 68, body mass index (BMI) between 18.5 and 22.9 kg/m^2^) and obesity (OB) (*n* = 68, BMI > 30.0 kg/m^2^). We then divided each group into 4 subgroups: 20–29, 30–39, 40–49, and 50–59 years. There were 34 participants in each subgroup (17 NW, 17 OB). The participants were included if they did not have any chronic diseases. Participants who were current smokers, abused alcohol, or took any medication or supplements were excluded. A written consent form was signed after receiving a verbal and written explanation before enrolment in the study.

Research manuscripts reporting large datasets that are deposited in a publicly available database should specify where the data have been deposited and provide the relevant accession numbers. If the accession numbers have not yet been obtained at the time of submission, please state that they will be provided during review. They must be provided prior to publication.

### 4.2. Research Design, Setting, and Protocol

This is a cross-sectional one-visit study which was performed during April 2019–September 2020 in Khon Kaen province, Thailand. After 12-h fasting, participants lay in the supine position, and heart rate (HR) and blood pressure (BP) were measured. Then, 16 mL blood was collected from the antecubital vein into four tubes: 1 mL blood in a sodium fluoride tube to measure fasting blood glucose (FBG); 9 mL blood in an ethylenediaminetetraacetic acid (EDTA) tube to measure lipid profiles, creatinine (Cr), haemoglobin A1c (HbA1c), malondialdehyde (MDA), and RTL (buffy coat); 3 mL blood in a serum separator tube to measure high-sensitivity C-reactive protein (hsCRP), insulin, and serum glutamic-pyruvic transaminase (SGPT); and 3 mL blood in a lithium heparin tube wrapped in aluminium foil. Perchloric acid (500 μL) was added to the 500 μL plasma aliquot from the last tube to precipitate proteins for vitamin C assessment. All tubes were centrifuged at 4 °C and 3000 g (TOMY-CAX-370, Tokyo, Japan) for 15 min, and the upper layer and buffy coat were transferred into a micro-centrifuge tube and stored at −80 °C until analysis. Then, expired air was collected to calculate RMR and RFO. After air collection, their aerobic capacity was measured. The laboratory temperature and humidity were 25 °C and 48%, respectively. Then, the participants provided demographic data, including education level, menstruation status, and marital status, and received daily dietary intake and physical activity (three days: 2 weekdays and 1 weekend day) forms to complete a week later.

### 4.3. Measurements of Physiological Characteristics

Resting HR and BP were measured by an automatic sphygmomanometer (SVM-7600 Nihon Kohden, Malaysia) on the upper right arm in supine position. FBG was measured by the glucose oxidase method (YSI Incorporated, Yellow Springs, OH, USA). Plasma lipid profile, Cr, and SGPT were measured by using Refloton Plus (Roach, Boehringer Mannheim, Germany). The coefficient of variation of measurement by using Refloton Plus was 0.1%. HbA1c was measured by a routine laboratory at Srinagarind Hospital (Cobas^®^ 6000 analyser series: Hitachi, Mannheim, Germany).

### 4.4. Measurements of Anthropometry and Body Composition

The anthropometry data were measured using a stadiometer with participants in a standing position with light clothes and bare feet (DETECTO, St. Webb City, MO, USA). Waist circumference was measured at the mid-point of the lower rib and iliac crest during full expiration, and hip circumference was measured at the widest point of the buttocks. Body fat (%) was measured by using a bioelectric impedance device (HBF-375 Omron, Kyoto, Japan). FFM was calculated by subtracting fat mass from body mass.

### 4.5. Measurement of RMR and RFO

Participants were in a supine position but were not allowed to sleep for 20 min. Then expired gas was collected to analyse carbon dioxide (CO_2_) and oxygen (O_2_) concentrations by indirect calorimetry (Oxycon CareFusion 234 GmbH, Hoechberg, Germany). Then, both gases were used to analyse the RMR by using the Weir equation [26], and RFO was calculated by using the Peronnet and Massicotte equation [32].

### 4.6. Measurement of Aerobic Capacity

We used maximum oxygen consumption ($\dot{V}$O_2max_) to indicate aerobic capacity, which was measured by a submaximal graded exercise test [33]. This test was performed on a cycle ergometer until the HR reached 85% of the maximum HR [34].

### 4.7. Measurements of Plasma Malondialdehyde, Vitamin C, and Inflammation

Lipid peroxidation was detected by the reaction of plasma malondialdehyde (MDA) with thiobarbituric acid (TBA) at low pH and high temperature to form a coloured complex, the MDA-TBA complex [35]. The plasma vitamin C concentration was measured by using Zhang’s method [36]. Inflammation was indicated by serum hsCRP which was measured by the laboratory at Srinagarind Hospital.

### 4.8. Telomere Length Measurement

Leukocyte DNA from buffy coat was used to measure RTL by using QuantStudio 6 Flex real-time polymerase chain reaction ABI StepOnePlus^TM^ system (Applied Biosystems, Grand Island, NY, USA) adapted from the method originally described by Cawthon [37] and previously reported [38] (Figure 4). Twofold serial dilutions of reference DNA ranging from 25 and 1.56 ng per 10 µL were run on every plate. All samples were run in triplicate. The RTL was shown as a proportion between telomeric repeat and single copy gene, 36b4 gene by T/S ratio. A tube contained PowerUp™ SYBR™ Green Master Mix (2×) (Applied Biosystems, Foster City, CA, USA), 12.5 ng/µL of DNA template, and 0.2 µM of telomere primers or 0.2 µM of single-copy gene primers. The thermal cycling profile for both telomeres and single copy genes starts with 95 °C incubation for 10 min, followed by 40 cycles of 15 sec at 95 °C and 1 min at 54 °C. The same reference DNA sample was used for each measurement to control interassay variability. The intra- and interassay coefficients of variation for TL measurement were 1.0% and 1.5%, respectively.

### 4.9. Statistical Analyses

All data are expressed as the mean ± standard deviation (SD) or median (interquartile range). The data were analysed by SPSS Statistics, version 28.0 (IBM Corp., Armonk, NY, USA). The normality of the distribution was tested by the Kolmogorov–Smirnov test. The mean difference in all data between the NW and OB groups was analysed by independent t test or Mann–Whitney U test, depending on the normality test. The Shapiro-Wilk test was used for normality test according to age subgroup. The normal distribution data were analysed by using analysis of variance (ANOVA); whereas the abnormal distribution data were analysed by using Kruskal Wallis test. A Generalized linear model (GLM) was used to analyse the association of RTL as a dependent variable and other continuous variables as independent variables. For visualization of unadjusted data, a heatmap generated by Pearson correlation (by the R package “corrplot” version 0.92) [39]. A significant result was indicated by a *p*-value less than 0.05.

## 5. Conclusions

This study shows that among 20- to 59-year-old, RTL had a negative association with RMR_AB_ and a positive association with aerobic capacity without any association with RFO in overall healthy sedentary Thai women. However, separate data according to obesity status both in the overall age and age subgroups showed different results. In the overall age, only NW participants had a negative association between RTL and all values of RMR; whereas OB participants had no association between RTL and all values of RMR. In the age subgroups, the OB participants showed that RTL was correlated with all values of RMR among 40- to 49-year-old and of aerobic capacity among 20- to 29-year-old. In contrast, NW participants revealed that RTL was only correlated with RMR, and aerobic capacity adjusted for FFM among 30- to 39-year-old. This may confirm that the correlations are dependent on age and obesity.

The mechanism underlying the longer RTL may be explained by the positive association with antioxidant indicated by vitamin C and negative association with anti-inflammatory effects indicated by hsCRP.

This study provides useful information for the management of promotion strategies for health-related physical fitness in women. Since the participants who had lower adiposity indices and greater aerobic capacity had longer RTL. Therefore, interventions that reduce obesity or increase aerobic capacity, such as low calorie diet and exercise training, may increase the participants’ longevity. Thus, it is worth encouraging women to control body weight and improve aerobic capacity. A further study proving these associations in male participants should be explored.

## Figures and Tables

**Figure 1 ijms-23-13336-f001:**
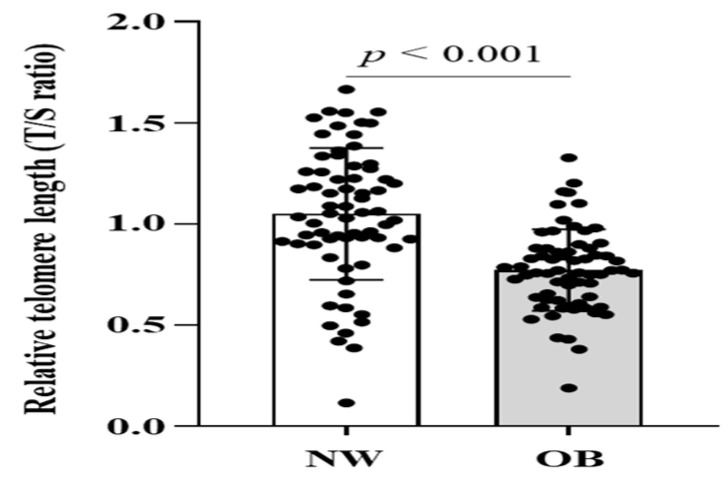
The blood leukocyte relative telomere length (T/S ratio) in participants with NW (*n* = 68) and OB (*n* = 66). The *p*-value was measured by independent *t*-test. Abbreviation: NW, normal weight; OB, obesity.

**Figure 2 ijms-23-13336-f002:**
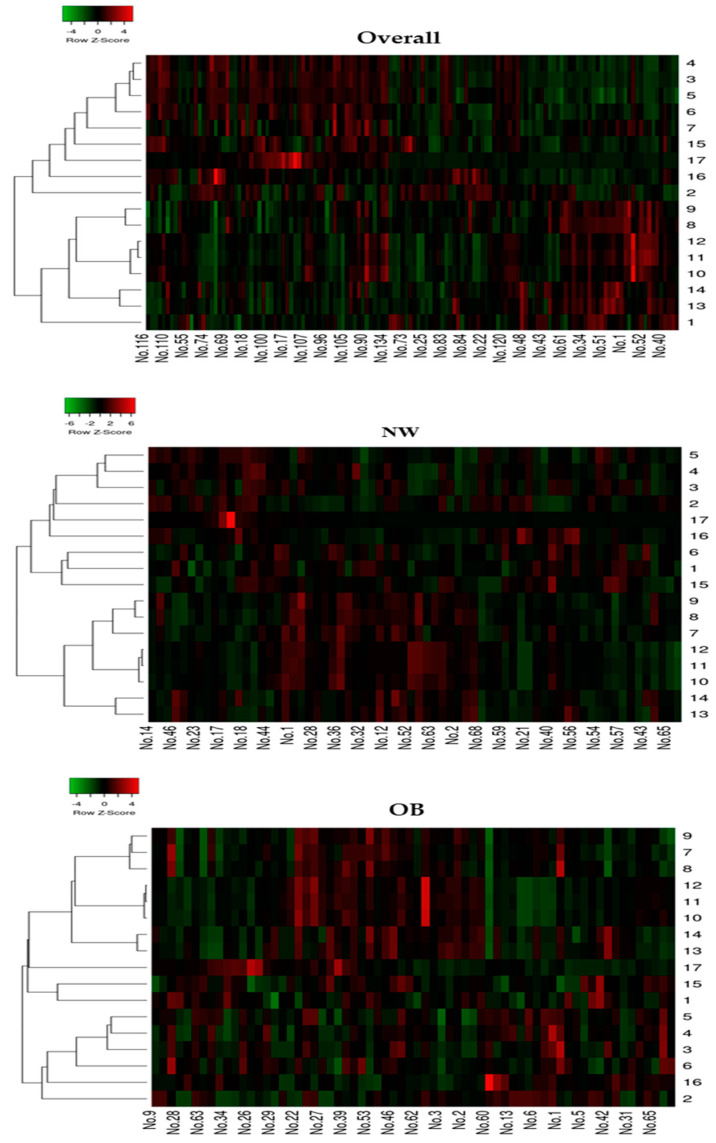
Heatmap of raw data (normalized to mean or median) and correlation of all the outcomes in the overall, NW, and OB groups. The colour red and green are representing the negative and positive correlations, respectively. **Symbols:** NW, normal weight; OB, obesity; Blood leukocyte relative telomere length = 1; age = 2; body mass index = 3; waist circumference = 4; body fat (%) = 5; fat-free mass = 6; RMR_AB_, absolute resting metabolic rate = 7; RMR_BM_, resting metabolic rate adjusted for body mass = 8; RMR_FFM_, resting metabolic rate adjusted for FFM = 9; RFO_AB_, absolute resting fat oxidation rate = 10; RFO_BM_, resting fat oxidation rate adjusted for body mass = 11; RFO_FFM_, resting fat oxidation rate adjusted for fat-free mass = 12; $\dot{V}$O_2max_ (mL/kg BM/min), maximum oxygen consumption adjusted for body mass = 13; $\dot{V}$O_2max_ (mL/kg FFM/min), maximum oxygen consumption adjusted for fat-free mass = 14; MDA, malondialdehyde = 16; hsCRP, high-sensitivity C-reactive protein = 17.

**Figure 3 ijms-23-13336-f003:**
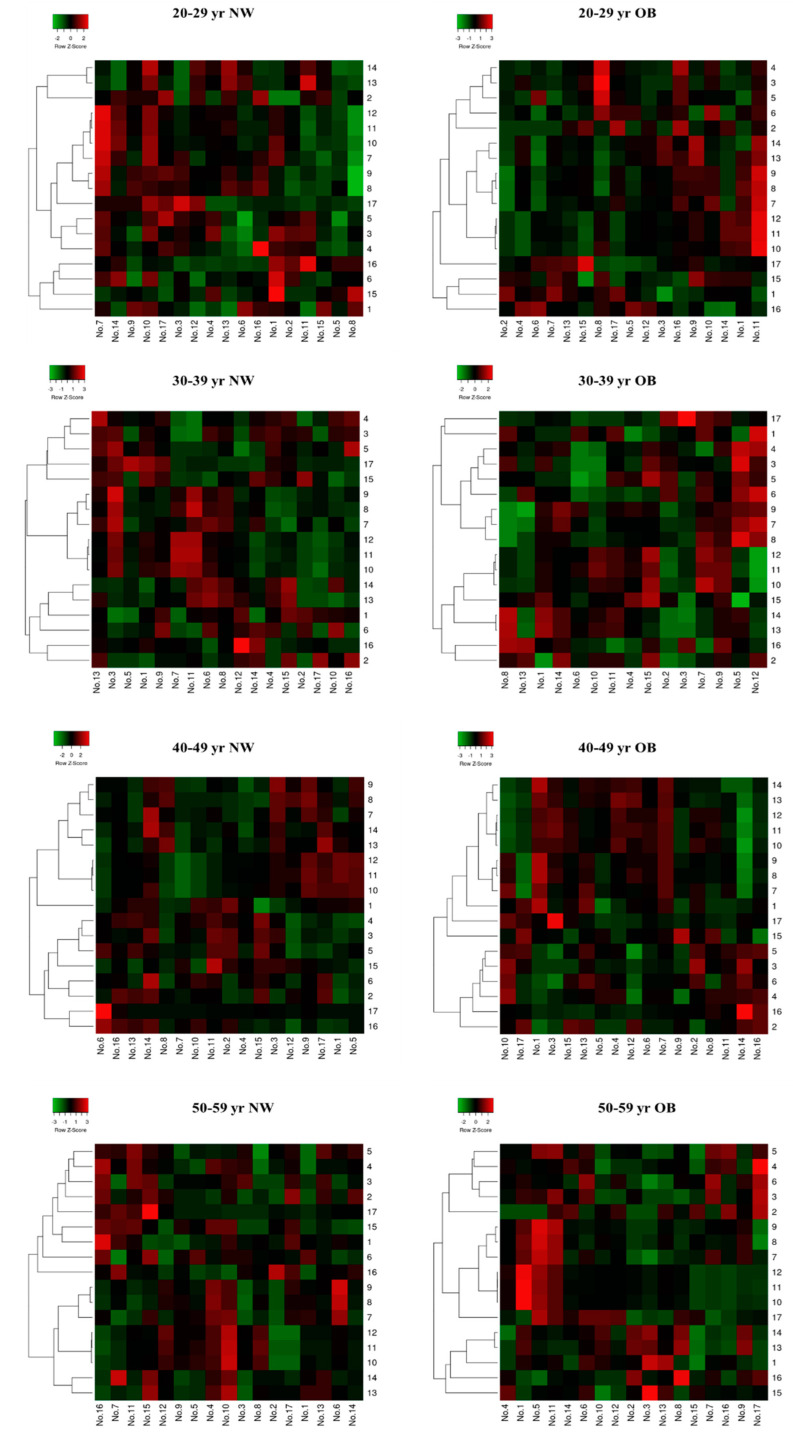
Heatmap of raw data (normalized to mean or median) and correlation of all the outcomes in the NW and OB groups. The colour red and green are representing the negative and positive correlations, respectively. **Symbol:** NW, normal weight; OB, obesity; blood leukocyte relative telomere length = 1; age = 2; body mass index = 3; waist circumference = 4; body fat (%) = 5; fat-free mass = 6; RMR_AB_, absolute resting metabolic rate = 7; RMR_BM_, resting metabolic rate adjusted for body mass = 8; RMR_FFM_, resting metabolic rate adjusted for FFM = 9; RFO_AB_, absolute resting fat oxidation rate = 10; RFO_BM_, resting fat oxidation rate adjusted for body mass = 11; RFO_FFM_, resting fat oxidation rate adjusted for fat-free mass = 12; $\dot{V}$O_2max_ (mL/kg BM/min), maximum oxygen consumption adjusted for body mass = 13; $\dot{V}$O_2max_ (mL/kg FFM/min), maximum oxygen consumption adjusted for fat-free mass = 14; MDA, malondialdehyde = 16; hsCRP, high-sensitivity C-reactive protein = 17.

**Figure 4 ijms-23-13336-f004:**
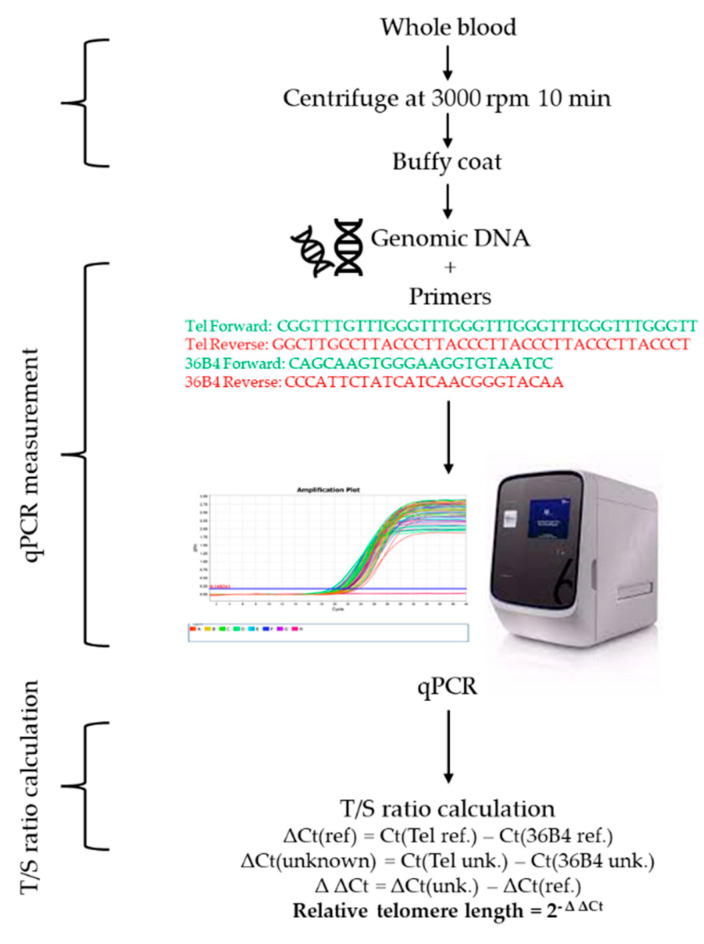
Diagram of qPCR blood leukocyte telomere length measurement adapted from previous study [37]. Abbreviation: DNA, Deoxyribonucleic acid; qPCR, QuantStudio 6 Flex real-time polymerase chain reaction.

**Table 1 ijms-23-13336-t001:** Metabolism, aerobic capacity, oxidative stress, inflammation of the participants.

	All (*n* = 134)	NW (*n* = 68)	OB (*n* = 66)	*p*-Value
RMR_AB_ (kcal/day)	1373.1 ± 271.8	1250.4 ± 194.1	1499.6 ± 283.5	<0.001
RMR_BM_ (kcal/kg BM/day)	21.9 ± 4.5	24.4 ± 3.6	19.4 ± 3.9	<0.001
RMR_FFM_ (kcal/kg FFM/day)	32.9 ± 5.2	34.2 ± 4.8	31.5 ± 5.3	0.002
RFO_AB_ (g/min)	0.056 ± 0.03	0.049 ± 0.02	0.063 ± 0.03	0.007
RFO_BM_ (g/kg BM/min)	0.0009 ± 0.0004	0.0010 ± 0.0005	0.0008 ± 0.0004	0.048
RFO_FFM_ (g/kg FFM/min)	0.0013 ± 0.0007	0.0013 ± 0.0006	0.0013 ± 0.0007	0.880
$\dot{V}$O_2max_ (mL/kg BM/min)	23.6 ± 5.9	26.3 ± 5.8	20.9 ± 4.6	<0.001
$\dot{V}$O_2max_ (mL/kg FFM/min)	35.5 ± 8.1	37.1 ± 8.4	33.8 ± 7.5	0.040
Plasma vitamin C (µmol/L)	51.4 ± 21.8	40.8 ± 16.7	62.4 ± 21.4	<0.001
Plasma MDA (µmol/mL)	6.8 (4.5–9.3)	5.6 (3.4–8.6)	8.3 (6.1–9.8)	0.001
Serum hsCRP (mg/L)	1.2 (0.4–3.1)	0.5 (0.2–1.2)	2.8 (1.4–4.1)	<0.001

Data are expressed as mean ± SD/median (interquartile range) based on normal distribution. Independent *t*-test or Mann-Whitney U test was used for the normally and abnormally distributed variables. The *p*-value compares between NW and OB group. Abbreviation: NW, normal weight; OB, obesity; BM, body mass; FFM, fat-free mass; RMR_AB_, absolute resting metabolic rate; RMR_BM_, resting metabolic rate adjusted for body mass; RMR_FFM_, resting metabolic rate adjusted for FFM; RFO_AB_, absolute resting fat oxidation rate; RFO_BM_, resting fat oxidation rate adjusted for body mass; RFO_FFM_, resting fat oxidation rate adjusted for fat-free mass; $\dot{V}$O_2max_ (mL/kg BM/min), maximum oxygen consumption adjusted for body mass; $\dot{V}$O_2max_ (mL/kg FFM/min), maximum oxygen consumption adjusted for fat-free mass; MDA, malondialdehyde; hsCRP, high-sensitivity C-reactive protein.

**Table 2 ijms-23-13336-t002:** Associations between RTL and age, body composition, metabolism, aerobic capacity, oxidative stress, and inflammation.

	Model	All (*n* = 134; 68 NW, 66OB)	*p* Value	NW (*n* = 68)	*p* Value	OB (*n* = 66)	*p*-Value
Age (yr)	Model1^#^	−0.004 (−0.009; 0.002)	0.164	−0.010 (−0.018; −0.001)	0.026	0.000 (−0.005; 0.005)	0.916
	Model3	−0.005 (−0.010; −0.008)	0.020	−0.012 (−0.018; −0.005)	0.001	−0.001 (−0.005; 0.004)	0.797
BMI (kg/m^2^)	Model1	−0.023 (−0.031; −0.014)	<0.001	0.021 (−0.033; 0.075)	0.443	0.003 (−0.019; 0.025)	0.778
	Model3	−0.021 (−0.296; −0.013)	<0.001	0.021 (−0.036; 0.078)	0.465	0.004 (−0.018; 0.025)	0.752
W (cm)	Model1	−0.008 (−0.012; −0.004)	<0.001	0.010 (−0.002; 0.022)	0.109	−0.002 (−0.009; 0.005)	0.599
	Model3	−0.008 (−0.012; −0.005)	<0.001	0.004 (−0.009; 0.017)	0.514	−0.002 (−0.009; 0.004)	0.513
BF (%)e	Model1	−0.021 (−0.030; −0.013)	<0.001	0.002 (−0.019; 0.024)	0.830	−0.003 (−0.029; 0.023)	0.818
	Model3	−0.023 (−0.031; −0.015)	<0.001	−0.013 (−0.034; 0.007)	0.210	−0.004 (−0.027; 0.020)	0.760
FFM (kg)	Model1	−0.016 (−0.024; −0.008)	<0.001	0.016 (−0.005; 0.037)	0.141	−0.003 (−0.016; 0.010)	0.667
	Model3	−0.015 (−0.019; −0.004)	0.002	0.028 (0.009; 0.047)	0.004	−0.001 (−0.013; 0.011)	0.916
RMR_AB_ (kcal/day)	Model1	−0.000 (−0.000; −0.000)	0.006	−0.000 (−0.001; 0.000)	0.239	−0.000 (−0.000; 0.000)	0.756
	Model2	−0.000 (−0.000; 0.000)	0.296	−0.000 (−0.001; −0.000)	0.049	−0.000 (−0.000; 0.000)	0.900
RMR_BM_ (kcal/kg BM/day)	Model1	0.009 (−0.002; 0.021)	0.115	−0.024 (−0.045; −0.004)	0.021	−0.001 (−0.014; 0.013)	0.903
	Model2	−0.008 (−0.020; 0.005)	0.230	−0.023 (−0.046; −0.000)	0.049	−0.001 (−0.015; 0.013)	0.877
RMR_FFM_ (kcal/kg FFM/day)	Model1	−0.001 (−0.011; 0.009)	0.841	−0.019 (−0.034; −0.003)	0.017	−0.001 (−0.011; 0.009)	0.901
	Model2	−0.007 (−0.016; 0.003)	0.169	−0.016 (−0.0327; −0.000)	0.045	−0.001 (−0.011; 0.010)	0.905
	Model3	−0.002 (−0.013; 0.008)	0.647	−0.018 (−0.034; −0.002)	0.027	0.000 (−0.010; 0.010)	0.998
RFO_AB_ (g/min)	Model1	−1.386 (−3.196; 0.424)	0.133	−1.038 (−4.616; 2.540)	0.570	−0.038 (−1.638; 1.562)	0.963
	Model2	−0.192 (−1.935; 1.551)	0.829	−1.196 (−4.806; 2.414)	0.516	−0.179 (−1.836; 1.478)	0.833
RFO_BM_ (g/kg BM/min)	Model1	14.6 (−106.5; 135.7)	0.813	−96.5 (−274.3; 81.3)	0.288	8.7 (−114.6; 132.0)	0.890
	Model2	−25.1 (−139.4; 89.3)	0.667	−67.8 (−250.4; 114.8)	0.467	−6.2 (−135.5; 123.2)	0.926
RFO_FFM_ (g/kg FFM/min)	Model1	−25.4 (−105.9; 55.1)	0.536	−65.2 (−194.0; 63.6)	0.321	2.8 (−72.2; 77.9)	0.941
	Model2	−14.0(−88.9; 61.0)	0.715	−46.6 (−176.9; 83.7)	0.484	−6.438 (−85.2; 72.3)	0.873
$\dot{V}$O_2max_ (mL/kg BM/min)	Model1	0.010 (0.001; 0.019)	0.026	−0.001 (−0.014; 0.013)	0.899	−0.003 (−0.015; 0.009)	0.640
	Model2	−0.000 (−0.010; 0.010)	0.982	0.001 (−0.013; 0.016)	0.858	−0.006 (−0.019; 0.008)	0.407
	Model3	0.013 (0.004; 0.022)	0.005	0.007 (−0.007; 0.021)	0.316	−0.002 (−0.013; 0.010)	0.787
$\dot{V}$O_2max_ (mL/kg FFM/min)	Model1	0.006 (−0.001; 0.012)	0.086	0.004 (−0.006; 0.013)	0.444	0.001 (−0.006; 0.008)	0.801
	Model2	0.003 (−0.003; 0.010)	0.332	0.004 (−0.005; 0.013)	0.394	−0.000 (−0.008; 0.008)	0.939
	Model3	0.007 (0.000; 0.013)	0.046	0.006 (−0.003; 0.016)	0.202	0.002 (−0.005; 0.009)	0.615
Plasma vitamin C (µmol/L)	Model1	−0.001 (−0.003; 0.002)	0.484	0.003 (−0.002; 0.008)	0.184	0.002 (−0.000; 0.005)	0.053
	Model2	0.003 (0.000; 0.005)	0.049	0.003 (−0.003; 0.008)	0.324	0.002 (−0.000; 0.005)	0.079
	Model3	−0.000 (−0.003; 0.002)	0.742	0.005 (0.000; 0.009)	0.048	0.002 (−0.000; 0.005)	0.060
	Model4	−0.000 (−0.003; 0.002)	0.892	0.004 (−0.000; 0.009)	0.066	0.003 (0.000; 0.005)	0.032
Plasma MDA (µmol/mL)	Model1	−0.005 (−0.017; 0.007)	0.418	0.010 (−0.009; 0.029)	0.301	−0.005 (−0.016; 0.007)	0.398
	Model2	0.001 (−0.011; 0.012)	0.913	0.006 (−0.013; 0.026)	0.528	−0.005 (−0.017; 0.007)	0.434
Serum hsCRP (mg/L)	Model1	−0.035 (−0.054; −0.015)	< 0.001	−0.025 (−0.063; 0.012)	0.190	−0.001 (−0.022; 0.020)	0.925
	Model2	−0.017 (−0.038; 0.004)	0.109	−0.031 (−0.071; 0.009)	0.132	−0.003 (−0.025; 0.019)	0.785
	Model3	−0.032 (−0.052; −0.012)	0.002	−0.032 (−0.071; 0.006)	0.096	0.000 (−0.019; 0.0196)	0.982

Data indicates beta-coefficient (95% CI) for the continuous variable in a GLM in NW and OB participants. Model 1^#^ = adjusted for energy expenditure, energy intake, and education level; Model 1 = adjusted for age, energy expenditure, energy intake, and education level; Model 2 = model1 add BMI, W circumference, body fat (%) and FFM; Model 3 = FBG; Model 4 = HOMA-IR. Abbreviation: RTL, blood leukocyte relative telomere length; NW, normal weight; OB, obese; BM, body mass; FFM, fat-free mass; RMR_AB_, absolute resting metabolic rate; RMR_BM_, resting metabolic rate adjusted for body mass; RMR_FFM_, resting metabolic rate adjusted for FFM; RFO_AB_, absolute resting fat oxidation rate; RFO_BM_, resting fat oxidation rate adjusted for body mass; RFO_FFM_, resting fat oxidation rate adjusted for fat-free mass; $\dot{V}$O_2max_ (mL/kg BM/min), maximum oxygen consumption adjusted for body mass; $\dot{V}$O_2max_ (mL/kg FFM/min), maximum oxygen consumption adjusted for fat-free mass; MDA, malondialdehyde; hsCRP, high-sensitivity C-reactive protein.

## Data Availability

Data are provided as requested.

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
