# Peer review of "Telomere Length Is Correlated with Resting Metabolic Rate and Aerobic Capacity in Women: A Cross-Sectional Study"

_ijms, 2022, doi:10.3390/ijms232113336_

Round 1

Reviewer 1 Report

The manuscript by  Rujira Nonsa-ard et al. entitled " Telomere length was associated with resting metabolic rate and aerobic capacity in women: a cross-sectional study" is an ambitious and very complex study. The increase in obesity worldwide has an important impact on health impairment and reduced quality of life, moreover being overweight or obese reduces life expectancy. The authors presented impressive research with an interesting idea and a massive number of data and comparisons. The structure of the article is logical and comprehensive, maybe besides, a few minor writing style issues.

However, the article has many minor and few major issues which should be explained or corrected.

Major issues:

(1) The number of data and comparisons is overwhelming for me, and maybe the readability of the manuscript could be improved. Perhaps some data could be moved to a supplementary file or the most important ones should be emphasized.

(2) Some issues should be discussed i.e. how long the participants were obese because a  telomere shortening is rather a long process

(3) In my opinion the conclusions could be extended or perhaps written in points

Minor issues

(1) Please specify what qPCR instrument was used…

(2) Lack Of Style In Communication in some parts i.e. line 20 to 23 an so on

(3) Aging or ageing which is non-american variant?

(4) Line  86 “that there were differences in the associations and mechanism between women with obesity and normal weight in each age subgroup. - Should be mechanisms?

(5)Line 104, 148, 406 the p value - Should be p-value?

(6) Line 118 to reveal the significant - should be significance?

(7) Line 292 fact that metabolic disorders are mostly occur  - should be occurred?

(8) Line 379 We used maximum oxygen consumELption (V̇O2max) to indicate aerobic capacity - EL?

My overall recommendation is to reconsider this manuscript after major revision.

Reviewer 2 Report

In this manuscript, Nonsa-ard et.al. investigated the association between relative telomere length with multiple indices of metabolism in sedentary woman. While the study design is well sorted, the manuscript needs some modifications as stated below:

1. Multiple areas of the manuscript lacks supporting evidences. e.g. Line 72 ("A G-strand..").

2. All the correlations should be shown via heatmap indicating the correlation values.

3. There is no experimental figures e.g. telomere length measurements.

Round 2

Reviewer 1 Report

The authors did a very good job and significantly improved the manuscript. Currently,  I have no more comments, thus I can recommend accepting the article in its present form. 

Author Response

Dear the reviewer 1

We appreciate your kind support for these sentences "The authors did a very good job and significantly improved the manuscript. Currently,  I have no more comments, thus I can recommend accepting the article in its present form." These sentences encourage us to work harder. 

Sincerely yours,

Naruemon Leelayuwat